# Barcodes as summary of objective functions' topology

## Abstract

We apply the canonical forms of gradient Morse complexes (barcodes) to explore topology of loss surfaces. We present a new algorithm for calculations of the objective function's barcodes of minima. Our experiments confirm two principal observations: (1) the barcodes of minima are located in a small lower part of the range of values of loss function of neural networks and (2) increase of the neural network's depth brings down the minima's barcodes. This has natural implications for the neural network learning and the ability to generalize.

## 1 Introduction

The learning via finding minima of objective functions is the principal strategy underlying majority of learning algorithms. For example, in Neural Network training, the objective function's input is model parameters (weights) and the objective function's output is the loss on training dataset. The graph of the loss function, often called **loss surface**, typically has complex structure (e.g. see loss surface visualisations by Li et al. (2018)): non-convexity, many local minima, flat regions, steep slopes. These obstacles harm exploration of the loss surface and complicate searching for optimal network weights.

The optimization of modern neural networks is based on the gradient descent algorithm. The global topological characteristics of the gradient vector field trajectories are captured by the Morse complex via decomposing the parameter space into cells of uniform flow, see Barannikov (1994); Le Roux et al. (2018) and references therein. The invariants of Morse complex called "canonical forms"(or barcodes) constitute the fundamental summary of the topology of the gradient vector field flow.

The "canonical forms", or barcodes, in this context are decompositions of the change of topology of the sublevel sets of objective function into simple "birth-death" phenomena of topological feautures of different dimensions.

The calculation of the barcodes for different functions constitutes the essence of the topological data analysis. The currently available software packages for the calculation of barcodes of functions, also called "sublevel persistence", are GUDHI, Dionysus, PHAT, and TDA package which incorporates all three previous packages B.T.Fasy et al. (2014). They are based on the algorithm, described in Barannikov (1994), see also appendix and e.g. Bauer et al. (2014) and references therein. This algorithm which has complexity of $O(n^3)$. These packages can currently handle calculations of barcodes for functions defined on a grid of up to $10^6$ points, and in dimensions two and three. Thus all current packages have the scalability issues.

We describe a new algorithm for computations of the barcodes of functions in lowest degree. Our algorithm works with functions defined on randomly sampled or specifically chosen **point clouds**. Point cloud based methods are known to work better than grid based methods in optimization related problems (Bergstra and Bengio (2012)). We also use the fact that the definition of the barcode of lowest degree can be reformulated in geometrical terms (see definition 1 in section 2). The previously known algorithms were based on the more algebraic approach as in definition 3. Our algorithm has complexity of $O(n \log(n))$. It was tested in dimensions up to 16 and with number of points of up to $10^8$.

**In this work**, we develop a methodology to describe the properties of the loss surface of the neural network via topological features of local minima.

We emphasize that the value of the objective function at the minimum can be viewed as only a part of its topological characteristic from the "canonical form" (barcode). The second half can be described as the value of objective function at the index-one saddle, which can be naturally associated with each local minimum.

The difference between the values of objective function at the associated index-one saddle and at the local minimum is a topological invariant of the minimum. For optimization algorithms this quantity measures, in particular, the obligatory penalty for moving from the given local minimum to a lower minimum.

**The main contributions of the paper are as follows:**

**Applying the one-to-one correspondence between local minima and 1-saddles to exploration of loss surfaces.** For each local minimum $p$ there is canonically defined 1-saddle $q$ (see Section 2). The 1-saddle associated with $p$ can be described as follows. The 1- saddle $q$ is precisely the point where the connected component of the sublevel set $\Theta_{f \leq c} = \{\theta \in \Theta \mid f(\theta) \leq c\}$ containing the minimum $p$ merges with another connected component of the sublevel set whose minimum is *lower*. This correspondence between the local minima and the 1-saddles, killing a connected component of $\Theta_{f \leq c}$, is one-to-one. The segment $[f(p), f(q)]$ is then the "canonical form" invariant attached to the minimum $p$. The set of all such segments is the barcode ("canonical form") of minima invariant of $f$. It is a robust topological invariant of objective function. It is invariant in particular under the action of homeomorphisms of $\Theta$. Full "canonical form" invariants give a concise summary of the topology of objective function and of the global structure of its gradient flow.

**Algorithm for calculations of the barcodes (canonical invariants) of minima.** We describe an algorithm for calculation of the canonical invariants of minima. The algorithm works with function's values on a a randomly sampled or specifically chosen set of points. The local minima give birth to clusters of points in sublevel sets. The algorithm works by looking at neighbors of each point with lower value of the function and deciding if this point belongs to the existing clusters, gives birth to a new cluster (minimum), or merges two or more clusters (index one saddle). A variant of the algorithm has complexity of $O(n \log(n))$, where $n$ is the cardinality of the set of points.

**Calculations confirming observations on behaviour of neural networks loss functions barcodes.** We calculate the canonical invariants (barcodes) of minima for small fully-connected neural networks of up to three hidden layers and verify that all segments of minima's barcode belong to a small lower part of the total range of loss function's values and that with the increase in the neural network depth the minima's barcodes descend lower.

The usefulness of our approach and algorithms is clearly not limited to the optimization problems. Our algorithm permits really fast computation of the canonical form invariants (persistence barcodes) of many functions which were not accessible until now. These sublevel persistence barcodes have been successfully applied in different discipline, to mention just a few: cognitive science (M. K. Chung and Kim (2009) ), cosmology (Sousbie et al. (2011)), see e.g. Pun et al. (2018) and references therein.

Our viewpoint should also have applications in chemistry and material science where 1-saddle points on potential energy landscapes correspond to transition states and minima are stable states corresponding to different materials or protein foldings (see e.g. Dellago et al. (2003), Oganov and Valle (2009)).

**The article is structured as follows.** First we describe three definitions of barcodes of minima. After that our algorithm for their calculation is described. In the last part we give examples of calculations, including the loss functions of simple neural nets.

## 2   TOPOLOGY OF LOSS SURFACES VIA CANONICAL FORM INVARIANTS

The "canonical form" invariants (barcodes) give a concise summary of topological features of functions (see Barannikov (1994), Le Roux et al. (2018) and references therein). These invariants describe a decomposition of the change of topology of the function into the finite sum of "birth"–"death" of elementary features. We propose to apply these invariants as a tool for exploring topology of loss surfaces.

In this work we concentrate on the part of these canonical form invariants, describing the "birth"–"death" phenomena of connected components of sublevel sets of the function.

However it should be stressed that this approach works similarly also for "almost minima", i.e. for the critical points (manifolds) of small indexes, which are often the terminal points of the optimization algorithms in very high dimensions.

We give three definitions of the "canonical form" invariants of minima.

DEFINITION 1: MERGING WITH CONNECTED COMPONENT OF A LOWER MINIMUM

The values of parameter $c$ at which the topology of sublevel set

$$\Theta_{f \leq c} = \{\theta \in \Theta \mid f(\theta) \leq c\}$$

changes are critical values of $f$.

Let $p$ be one of minima of $f$. When $c$ increases from $f(p) - \epsilon$ to $f(p) + \epsilon$, a new connected component of the set $\Theta_{f \leq c}$ is born (see fig 1a, the connected components $S_1, S_2, S_3$ of sublevel set are born at the blue, green and red minima correspondingly.

If $p$ is a minimum, which is not global, then, when $c$ is increased, the connected component of $\Theta_{f \leq c}$ born at $p$ merges with a connected component born at a *lower* minimum. Let $q$ is the merging point where this happens. The intersection of the set $\Theta_{f < f(q)}$ with any small neighborhood of $q$ has two connected components. This is the index-one saddle $q$ associated with $p$.

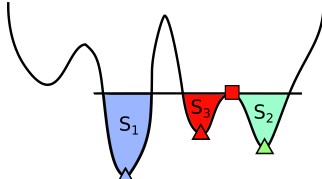 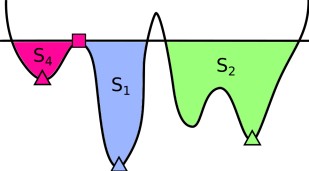 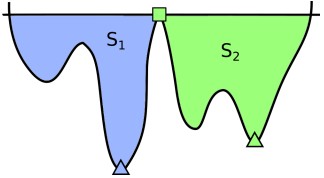

(a) "Death" of the connected component $S_3$. The connected component $S_3$ of sublevel set merges with connected component $S_2$ at red saddle, red saddle is associated with the red minimum.

(b) "Death" of the connected component $S_4$. The connected component $S_4$ of sublevel set merges with connected component $S_1$ at violet saddle, violet saddle is associated with the violet minimum

(c) "Death" of the connected component $S_2$. The connected component $S_2$ of sublevel set merges with connected component $S_1$ at green saddle, green saddle is associated with the green minimum.

Figure 1: Merging of connected components of sublevel sets at saddles. Note that the green saddle is associated with the green minimum which is separated by another minimum from the green saddle.

Also these two subsets of small neighborhood of $q$ belong to two different connected components of the whole set $\Theta_{f < f(q)}$. The 1-saddles of this type are called "+" ("plus") or "death" type. The described correspondence between local minima and 1-saddles of this type is one-to-one.

In a similar way, the 1-saddle $q$ associated with $p$ can be described also as follows.

**Proposition 2.1.** *Consider various paths $\gamma$ starting from the local minimum $p$ and going to a lower minimum. Let $m_\gamma \in \Theta$ is the maximum of the restriction of $f$ to such path $\gamma$. Then 1-saddle $q$ which is paired with the local minimum $p$ is the minimum over the set of all such paths $\gamma$ of the maxima $m_\gamma$:*

$$q = arg \min_{\substack{\gamma:[0,1] \to \Theta \\ \gamma(0)=p,\, f(\gamma(1)) < f(p)}} \left[ \max_t f\big(\gamma(t)\big) \right]$$

DEFINITION 2: NEW MINIMUM ON CONNECTED COMPONENTS OF SUBLEVEL SETS

The correspondence in the opposite direction can be described analogously. Let $q$ is a 1-saddle point of such type that the two branches of the set $\Theta_{f \leq f(q) - \epsilon}$ near $q$ belong to two different connected components of $\Theta_{f \leq f(q) - \epsilon}$. A new connected component of the set $\Theta_{f \leq c}$ is formed when $c$ decreases from $f(q) + \epsilon$ to $f(q) - \epsilon$. The restriction of $f$ to each of the two connected components has its global minimum.

**Proposition 2.2.** *Given a 1-saddle $q$, the minimum $p$ which is paired with $q$ is the new minimum of $f$ on the connected component of the set $\Theta_{f \leq c}$ which is formed when $c$ decreases from $f(q) + \epsilon$ to $f(q) - \epsilon$. $\square$*

The two branches of the set $\Theta_{f \leq f(q) - \epsilon}$ near $q$ can also belong to the same connected components of this set. Then such saddle is of "birth" type and it is naturally paired with index-two saddle of "death" type (see theorem 2.3).

DEFINITION 3: INVARIANTS OF FILTERED COMPLEXES

Chain complex is the algebraic counterpart of intuitive idea representing complicated geometric objects as a decomposition into simple pieces. It converts such a decomposition into a collection of vector spaces and linear maps.

A chain complex $(C_*, \partial_*)$ is a sequence of finite-dimensional $k$-vector spaces and linear operators

$$\to C_{j+1} \overset{\partial_{j+1}}{\to} C_j \overset{\partial_j}{\to} C_{j-1} \to \ldots \to C_0,$$

which satisfy

$$\partial_j \circ \partial_{j+1} = 0.$$

The $j-$th homology of the chain complex $(C_*, \partial_*)$ is the quotient

$$H_j = \ker(\partial_j) / \operatorname{im}(\partial_{j+1}).$$

A chain complex $C_*$ is called $\mathbb{R}-$filtered if $C_*$ is equipped with an increasing sequence of sub-complexes ($\mathbb{R}-$filtration) $F_{s_1} C_* \subset F_{s_2} C_* \subset \ldots \subset F_{s_{max}} C_* = C_*$, indexed by a finite set of real numbers $s_1 < s_2 < \ldots < s_{\max}$.

**Theorem 2.3.** *(Barannikov (1994)) Any $\mathbb{R}-$filtered chain complex $C_*$ can be brought by a linear transformation preserving the filtration to "canonical form", a canonically defined direct sum of $\mathbb{R}-$filtered complexes of two types: one-dimensional complexes with trivial differential $\partial_j(e_i) = 0$ and two-dimensional complexes with trivial homology $\partial_j(e_{i_2}) = e_{i_1}$. The resulting canonical form is uniquely determined.*

The full barcode is a visualization of the decomposition of an $\mathbb{R}-$filtered complexes according to the theorem 2.3. Each filtered 2-dimensional complex with trivial homology $\partial_j(e_{i_2}) = e_{i_1}$, $\langle e_{i_1} \rangle = F_{\leq s_1}, \langle e_{i_1}, e_{i_2} \rangle = F_{\leq s_2}$ describes a topological feature in dimension $j$ which is "born" at $s_1$ and which "dies" at $s_2$. It is represented by segment $[s_1, s_2]$ in the degree-$j$ barcode. And each filtered 1-dimensional complex with trivial differential, $\partial_j e_i = 0$, $\langle e_i \rangle = F_{\leq r}$ describes a topological feature in dimension $j$ which is "born" at $r$ and never "dies". It is represented by the half-line $[r, +\infty[$ in the degree-$j$ barcode.

The proof of the theorem is given in Appendix. Essentially, one can bring an $\mathbb{R}-$filtered complex to the required canonical form by induction, starting from the lowest basis elements of degree one, in such a way that the manipulation of degree $j$ basis elements does not destroy the canonical form in degree $j - 1$ and in lower filtration pieces in degree $j$.

Let $f : \Theta \to \mathbb{R}$ is smooth, or more generally, piece-wise smooth continuous function such that the sublevel sets $\Theta_{f \leq c} = \{\theta \in \Theta \mid f(\theta) \leq c\}$ are compact.

One filtered complex naturally associated with function $f$ and such that the subcomplexes $F_s C_*$ compute the homology of sublevel sets $\Theta_{f \leq s}$ is the gradient (Morse) complex, see e.g. Barannikov (1994); Le Peutrec et al. (2013) and references therein. Without loss of generality the function $f$ can be assumed smooth here, otherwise one can always replace $f$ by its smoothing. By adding a small perturbation such as a regularization term we can also assume that critical points of $f$ are non-degenerate.

The generators of the gradient (Morse) complex correspond to the critical points of $f$. The differential is defined by counting gradient trajectories between critical points when their number is finite.

The canonical form of the gradient (Morse) complex describes a decomposition of the gradient flow associated with $f$ into standard simple pieces.

Let $p$ be a minimum, which is not a global minimum. Then the generator corresponding to $p$ represents trivial homology class in the canonical form, since the homology class of its connected component is already represented by the global minimum. Then $p$ is the lower generator of a two-dimensional complex with trivial homology in the canonical form. I.e. $p$ is paired with an index-one saddle $q$ in the canonical form. The segment $[f(p), f(q)]$ is then the canonical invariant (barcode) corresponding to the minimum $p$.

The full canonical form of the gradient (Morse) complex of all indexes is a summary of global structure of the objective function's gradient flow.

The total number of different topological features in sublevel sets $\Theta_{f \leq c}$ of the objective function can be read immediately from the barcode. Namely the number of intersections of horizontal line at level $c$ with segments in the index $j$ barcode gives the number of independent topological features of dimension $j$ in $\Theta_{f \leq c}$.

The description of the barcode of minima on manifold $\Theta$ with nonempty boundary $\partial \Theta$ is modified in the following way. A connected component can be also born at a local minimum of restriction of $f$ to the boundary $f \mid_{\partial \Theta}$, if $\mathrm{grad} f$ is pointed inside manifold $\Theta$. The merging of two connected components can also happen at an index-one critical point of $f \mid_{\partial \Theta}$, if $\mathrm{grad} f$ is pointed inside $\Theta$.

## 3 AN ALGORITHM FOR CALCULATION OF BARCODES OF MINIMA

In this section we describe the developed algorithm for calculation of the canonical form invariants of local minima. The computation exploits the first definition of barcodes (see Section 2), which is based on the evolution on the connected components of the sublevel sets.

To analyse the surface of the given function $f : \Theta \to \mathbb{R}$, we first build its **approximation** by finite graph-based construction. To do this, we consider a random subset of points $\{\theta_1, \ldots, \theta_N\} \in \Theta$ and build a graph with these points as vertices. The edges connect **close** points. Thus, for every vertex $\theta_n$, by comparing $f(\theta_n)$ with $f(\theta_{n'})$ for neighbors $\theta_{n'}$ of $\theta_n$, we are able to understand the local topology near the point $\theta_n$. At the same time, connected componenets of sublevel sets $\Theta_{f \leq c} = \{\theta \in \Theta \mid f(\theta) \leq c\}$ will naturally correspond to connected components of the subgraph on point $\theta_n$, such that $f(\theta_n) \leq c$.[1]

Two technical details here are the choice of points $\theta_n$ and the definition of closeness, i.e. when to connect points by an edge. In our experiments, we sample points uniformly from some rectangular box of interest. To add edges, we compute the oriented $k$-Nearest Neighbor Graph on the given points and then drop the orientation of edges. Thus, every node in the obtained graph has a degree in $[k, 2k]$. In all our experiments we use $k = 2D$, where $D$ is the dimension of $f$'s input.

Next we describe our **algorithm**, which computes barcodes of a function from its graph-based approximation described above. The key idea is to monitor the evolution of the connected components of the sublevel sets of the graph, i.e. sets $\Theta_c = \{\theta_n \mid f(\theta_n) \leq c\}$ for increasing $c$.

For simplicity we assume that points $\theta$ are ordered w.r.t. the value of function $f$, i.e. for $n < n'$ we have $f(\theta_n) < f(\theta_{n'})$. In this case we are interested in the evolution of connected components throughout the process sequential adding of vertices $\theta_1, \theta_2, \ldots, \theta_N$ to graph, starting from an empty graph. We denote the subgraph on vertices $\theta_1, \ldots, \theta_n$ by $\Theta_n$. When we add new vertex $\theta_{n+1}$ to $\theta_n$, there are three possibilities for connected componenets to evolve:

1. Vertex $\theta_{n+1}$ has zero degree in $\Theta_{n+1}$. This means that $\theta_{n+1}$ is a local minimum of $f$ and it forms a new connected component in the sublevel set.

2. All the neighbors of $\theta_{n+1}$ in $\Theta_{n+1}$ belong to one connected component in $\Theta_n$.

3. All the neighbors of $\theta_{n+1}$ in $\Theta_{n+1}$ belong to $\geq 2$ connected components $s_1, s_2, \ldots, s_K \subset \Theta_n$. Thus, all these components will form a single connected component in $\Theta_{n+1}$.

---

[1] In fact we build a filtered simplicial complex, which approximates the function plot. Its degree zero chains are spanned by the points $\theta_n$, and degree one chains are spanned by the edges between close pairs of points.

---

**Algorithm 1:** Barcodes of minima computation for function on a graph.

---

**Input** : Connected undirected graph $G = (V, E)$; function $f$ on graph vertices.
**Output :** Barcodes: a list of "birth"-"death" pairs.
$S \leftarrow \{\}$;
$f^* \leftarrow \min f(\theta)$ for $\theta \in V$;
Barcodes $\leftarrow [(f^*, \infty)]$;
**for** $\theta \in V$ *in increasing order of* $f(\theta)$ **do**
  $S' \leftarrow \{s \in S \mid \exists \theta' \in s$ such that $(\theta, \theta') \in E$ and $f(\theta) > f(\theta')\}$;
  **if** $S' = \emptyset$ **then**
    $S \leftarrow S \cup \{\{\theta\}\}$;
  **else**
    $f^* \leftarrow \min f(\theta')$ for $\theta' \in \bigsqcup\limits_{s \in S'} s$;
    **for** $s \in S'$ **do**
      $f^s \leftarrow \min f(\theta')$ for $\theta' \in s$;
      **if** $f^s \neq f^*$ **then**
        Barcodes $\leftarrow$ Barcodes $\cup \{(f^s, f(\theta))\}$;
    **end**
    $s_{\text{new}} \leftarrow \left( \bigsqcup\limits_{s \in S'} s \right) \sqcup \{\theta\}$;
    $S \leftarrow (S \setminus S') \sqcup \{s_{\text{new}}\}$;
  **end**
  **return** Barcodes
**end**

---

In the third case, according to definition 1 of Section 2 the point $\theta_{n+1}$ is a 1-saddle point. Thus, one of the components $s_k$ swallows the rest. This is the component which has the lowest minimal value. For other components,[2] this gives their barcodes: for $s_k$ the birth-death pair is $\left( \min\limits_{\theta \in s_k} f(\theta); f(\theta_{n+1}) \right)$.

We summarize the procedure in the following algorithm 1. Note that we assume that the input graph is connected (otherwise the algorithm can be run on separate connected components).

In the practical implementation of the algorithm, we precompute the values of function $f$ at all the vertices of $G$. Besides that, we use the disjoint set data structure to store and merge connected components during the process. We also keep and update the global minima in each component. We did not include these tricks into the algorithm's pseuso-code in order to keep it simple.

The resulting complexity of the algorithm is $O(N \log N)$ in the number of points. Here it is important to note that the procedure of graph creation may be itself time-consuming. In our case, the most time consuming operation is nearest neighbor search. In our code, we used efficient **HNSW Algorithm** for aproximate NN search by Malkov and Yashunin (2018).

## 4 EXPERIMENTS

In this section we apply our algorithm to describing the surfaces of functions. In Subsection 4.1 we apply the algorithm to **toy visual examples**. In Subsection 4.2 we apply the algorithm to analyse the loss surfaces of **small neural networks**.

### 4.1 TOY FUNCTIONS

In this subsection we demonstrate the application of the algorithm to simple toy functions $f : \mathbb{R}^D \to \mathbb{R}$. For $D \in \{1, 2\}$ we consider three following functions:

---

[2]Typically it merges **two** connected components of $\Theta_n$. However, due to noise and non-dense approximation of function by graph in high-dimensional spaces, it may happen that it merges more than two connected components.

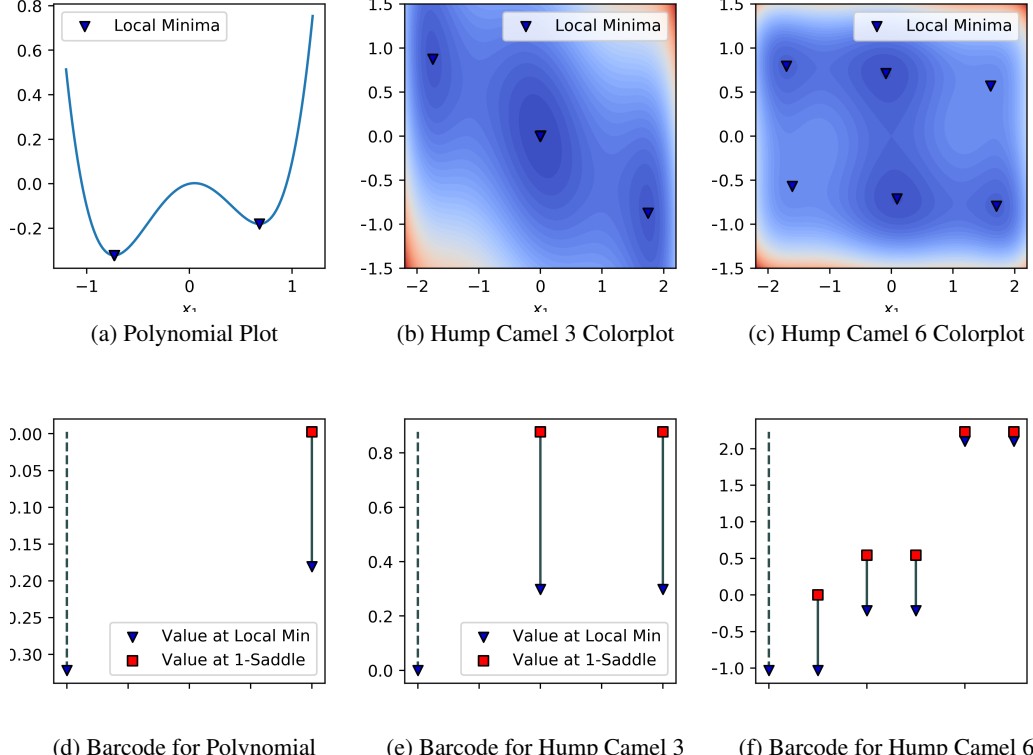

Figure 2: Plots (first row) and the coresponding barcodes (second row) for Polynomial of Degree 4, Hump Camel 3, Hump Camel 6 functions respectively.

1. **Polynomial** of a single variable of degree 4 with 2 local minima (see Fig. 2a):

$$f(\theta_1) = \theta_1^4 - \theta_1^2 + \frac{\theta_1}{10} \qquad (1)$$

2. **Camel function with 3 humps**, i.e. 3 local minima (see Fig. 2b):

$$f(\theta_1, \theta_2) = (2 - 1.05\theta_1^2 + \theta_1^4/6)\theta_1^2 + \theta_1\theta_2 + \theta_2^2 \qquad (2)$$

3. **Camel function with 6 humps**, i.e. 6 local minima (see Fig. 2c):

$$f(\theta_1, \theta_2) = (4 - 2.1\theta_1^2 + \theta_1^4/3)\theta_1^2 + \theta_1\theta_2 + (-4 + 4\theta_2^2)\theta_2^2 \qquad (3)$$

Function plots with their corresponding barcodes of minima are given in Figure 2. The barcode of the global minimum is represented by the dashed half-line which goes to infinity.

## 4.2 TOPOLOGY OF NEURAL NETWORK LOSS FUNCTION

In this section we compute and analyse barcodes of small fully connected neural networks with up to three hidden layers.

For several architectures of the neural networks many results on the loss surface and its local minima are known (see e.g. Kawaguchi (2016) Gori and Tesi (1992) and references therein). Different geometrical and topological properties of loss surfaces were studied in Cao et al. (2017); Yi et al. (2019); Chaudhari et al. (2017); Dinh et al. (2017).

There is no ground truth on how should the **best loss surface** of a neural network looks like. Nevertheless, there exists many common opinions on this topic. First of all, from practical optimization point of view, the desired local (or global) minima should be easily reached via basic training methods such

as Stochastic Gradient Descent, see Ruder (2016). Usually this requires more-or-less stable slopes of the surface to prevent instabilities such as gradient explosions or vanishing gradients. Secondly, the value of obtained minimum is typically desired to be close to global, i.e. attain smallest training error. Thirdly, from the generalization point of view, such minima are required to provide small loss on the testing set. Although in general it is assumed that the good local optimum is the one that is flat, some recent development provide completely contrary arguments and examples, e.g. sharp minima that generalize well.

Besides the optimization of the weights for a given architecture, neural network training implies also a choice of the architecture of the network, as well as the loss function to be used for training. In fact, it is the choice of the architecture and the loss function that determines the shape of the loss surface. Thus, proper selection of the network's architecture may simplify the loss surface and lead to potential improvements in the weight optimization procedure.

We have analyzed very tiny neural networks. However our method permits full exploration of the loss surface as opposed to stochastical exploration of higher-dimensional loss surfaces. Let us emphasize that even from practical point of view it is important to understand first the behavior of barcodes in simplest examples where all hyper-parameters optimization schemes can be easily turned off. For every analysed neural network the objective function is its mean squared error for predicting (randomly selected) function $g : [-\pi, \pi] \to \mathbb{R}$ given by

$$g(x) = 0.31 \cdot \sin(-x) - 0.72 \cdot \sin(-2x) - 0.21 \cdot \cos(x) + 0.89 \cdot \cos(2x)$$

plus $l_2-$regularization. The error is computed for prediction on uniformly distributed inputs $x \in \{-\pi + \frac{2\pi}{100}k \mid k = 0, 1, \ldots, 100\}$.

The neural networks considered were fully connected one-hidden layer with 2, 3 and 4 neurons, two-hidden layers with 2x2, 3x2 and 3x3 neurons, and three hidden layers with 2x2x2 and 3x2x2 neurons. We have calculated the barcodes of the loss functions on the hyper-cubical sets $\Theta$ which were chosen based on the typical range of parameters of minima. The results are as shown in Figure 3.

We summarize our findings into two main observations:

1. the barcodes are located in tiny lower part of the range of values; typically the maximum value of the function was around 200, and the saddles paired with minima lie below 1;

2. with the increase of the neural network depth the barcodes descend lower.

For example the upper bounds of barcodes of one-layer (2) net are in range $[0.55, 0.65]$, two-layer $(2 \times 2)$ net in range $[0.35, 0.45]$, and three-layer $(2 \times 2 \times 2)$ net in range $[0.1, 0.3]$.

## 5 CONCLUSION

In this work we have introduced a methodology for analysing the plots of functions, in particular, loss surfaces of neural networks. The methodology is based on computing topological invariants called canonical forms or barcodes.

To compute barcodes we used a graph-based construction which approximates the function plot. Then we apply the algorithm we developed to compute the barcodes of minima on the graph. Our experimental results of computing barcodes for small neural networks lead to two principal observations.

First all barcodes sit in a tiny lower part of the total function's range. Secondly, with increase of the depth of neural network the barcodes descend lower. From the practical point of view, this means that gradient descent optimization cannot stuck in high local minima, and it is also not difficult to get from one local minimum to another (with smaller value) during learning.

The method we developed has several further research directions. Although we tested the method on small neural networks, it is possible to apply it to large-scale modern neural networks such as convolutional networks (i.e. ResNet, VGG, AlexNet, U-Net, see Alom et al. (2018)) for image-processing based tasks. However, in this case the graph-based approximation we use requires wise choice of representative graph vertices, which is a hardcore in high-dimensional spaces (dense filling

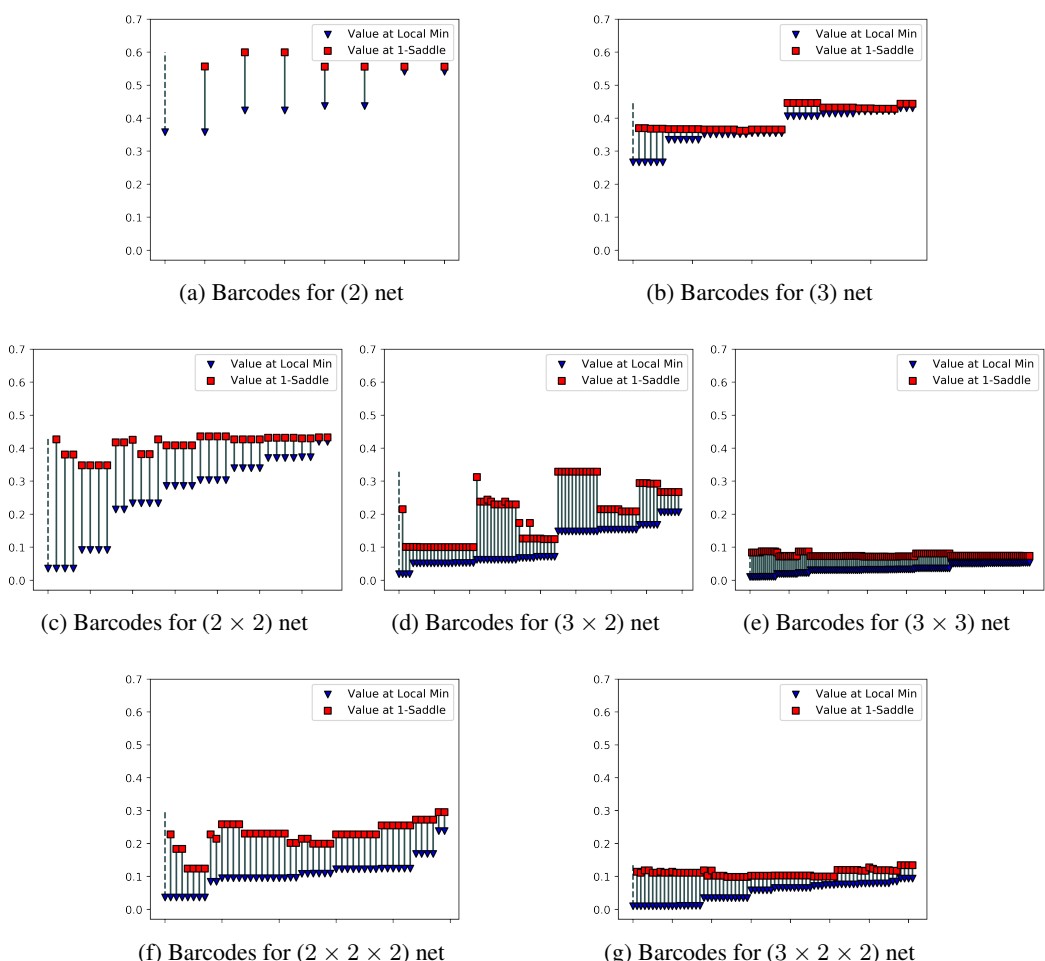

(a) Barcodes for (2) net

(b) Barcodes for (3) net

(c) Barcodes for (2 × 2) net

(d) Barcodes for (3 × 2) net

(e) Barcodes for (3 × 3) net

(f) Barcodes for (2 × 2 × 2) net

(g) Barcodes for (3 × 2 × 2) net

Figure 3: Barcodes of different neural network loss surfaces.

of area by points is computationally intractable). Another direction is to study the connections between the barcode of local minima and the generalization properties of given minimum and of neural network. There are clearly also connections, deserving further investigation, between the barcodes of minima and results concerning the rate of convergency during learning of neural networks.

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

## APPENDIX

### 5.1 PROOF OF THE THEOREM 2.3

The theorem is similar in spirit to the bringing a quadratic form to a sum of squares.

*Proof.* (Barannikov (1994)) Let's choose a basis in the vector spaces $C_n$ compatible with the filtration, so that each subspace $F_r C_n$ is the span $\left\langle e_1^{(n)}, \ldots, e_{i_r}^{(n)} \right\rangle$.

Let $\partial e_l^{(n)}$ has the required form for $n = j$ and $l \leq i$, or $n < j$ and all $l$. I.e. either $\partial e_l^{(n)} = 0$ or $\partial e_l^{(n)} = e_{m(l)}^{(n-1)}$, where $m(l) \neq m(l')$ for $l \neq l'$.

Let

$$\partial e_{i+1}^{(j)} = \sum_k e_k^{(j-1)} \alpha_k.$$

Let's move all the terms with $e_k^{(j-1)} = \partial e_q^j$, $q \leq i$, from the right to the left side. We get

$$\partial (e_{i+1}^{(j)} - \sum_{q \leq i} e_q^{(j)} \alpha_{k(q)}) = \sum_k e_k^{(j-1)} \beta_k$$

If $\beta_k = 0$ for all $k$, then define

$$\tilde{e}_{i+1}^{(j)} = e_{i+1}^{(j)} - \sum_{q \leq i} e_q^{(j)} \alpha_{k(q)},$$

so that

$$\partial \tilde{e}_{i+1}^{(j)} = 0,$$

and $\partial e_l^{(n)}$ has the required form for $l \leq i+1$ and $n = j$, and for $n < j$ and all $l$.

Otherwise let $k_0$ be the maximal $k$ with $\beta_k \neq 0$. Then

$$\partial (e_{i+1}^{(j)} - \sum_{q \leq i} e_q^{(j)} \alpha_{k(q)}) = e_{k_0}^{(j-1)} \beta_{k_0} + \sum_{k < k_0} e_k^{(j-1)} \beta_k, \ \beta_{k_0} \neq 0.$$

Define

$$\tilde{e}_{i+1}^{(j)} = \left( e_{i+1}^{(j)} - \sum_{q \leq i} e_q^{(j)} \alpha_{k(q)} \right) / \beta_{k_0}, \ \tilde{e}_{k_0}^{(j-1)} = e_{k_0}^{(j-1)} + \sum_{k < k_0} e_k^{(j-1)} \beta_k / \beta_{k_0}.$$

Then

$$\partial \tilde{e}_{i+1}^{(j)} = \tilde{e}_{k_0}^{(j-1)}$$

and for $n = j$ and $l \leq i+1$, or $n < j$ and all $l$, $\partial e_l^{(n)}$ has the required form. If the complex has been reduced to "canonical form" on subcomplex $\oplus_{n \leq j} C_n$, then reduce similarly $\partial e_1^{(j+1)}$ and so on.

Uniqueness of the canonical form follows essentially from the uniqueness at each previous step. Let $\left\{ a_i^{(j)} \right\}$, $\left\{ b_i^{(j)} = \sum_{k \leq i} a_k^{(j)} \alpha_k \right\}$, be two bases of $C_*$ for two different canonical forms. Assume that for all indexes $p < j$ and all $n$, and $p = j$ and $n \leq i$ the canonical forms agree. Let $\partial a_{i+1}^{(j)} = a_m^{(j-1)}$ and $\partial b_{i+1}^{(j)} = b_l^{(j-1)}$ with $m > l$.

It follows that

$$\partial \left( \sum_{k \leq i+1} a_k^{(j)} \alpha_k \right) = \sum_{n \leq l} a_n^{(j-1)} \beta_n,$$

where $\alpha_{i+1} \neq 0$, $\beta_l \neq 0$. Therefore

$$\partial a_{i+1}^{(j)} = \sum_{n \leq l} a_n^{(j-1)} \beta_n / \alpha_{i+1} - \sum_{k \leq i} \partial a_k^{(j)} \alpha_k / \alpha_{i+1}.$$

On the other hand $\partial a_{i+1}^{(j)} = a_m^{(j-1)}$, with $m > l$, and $\partial a_k^{(j)}$ for $k \leq i$ are either zero or some basis elements $a_n^{(j-1)}$ different from $a_m^{(j-1)}$. This gives a contradiction.

Similarly if $\partial b_{i+1}^{(j)} = 0$, then

$$\partial a_{i+1}^{(j)} = - \sum_{k \leq i} \partial a_k^{(j)} \alpha_k / \alpha_{i+1}$$

which again gives a contradiction by the same arguments. Therefore the canonical forms must agree for $p = j$ and $n = i+1$ also. $\qquad \square$

