# OpenReview forum: "Barcodes as summary of objective functions' topology"
_ICLR.cc/2020/Conference — Reject_

### Official Review · AnonReviewer2 · 2019-10-21
**Official Blind Review #2**

**Rating:** 1

**Review:**

This work is focused on topological characterization of target surfaces of optimization objectives (i.e. loss functions) by computing so called barcodes, which are lists of pairs of local minima and their connected saddle points. The authors claim that the barcodes constitute a representation of target objectives that is invariant under homeomorphisms of input to the objectives. The authors present an algorithm for computing the barcodes from graph-based representation of a surface, and present barcodes computed on toy examples in numerical analysis.

In my opinion, the main contribution of the work i.e. creation of barcodes is based on a rather trivial idea. The concept of characterizing optimization objectives through pairs of local minima and one-index saddle points is straightforward given that one can (thoroughly if not exhaustively) compute them in a computationally feasible manner; this is however hardy the case in any realistic scenario. I therefore struggle to see how the idea can be practically significant. Maybe the authors can put more emphasis on the theoretical aspect of their work, which is about the invariance nature of barcodes. They can for instance demonstrate how one can exploit the invariance property of barcodes for parameter optimization.

The authors can consider application of their work to hyper-parameter optimization, which is usually low-dimensional and one can also compare with other approaches such as Gaussian processes or other Bayesian methodologies.

In numerical experiments, for the toy task solved using neural network I don't find it very surprising that the barcodes descend lower as the capacity of the network is increased. Can the authors further clarify why it is a significant finding for them?

**Experience Assessment:**

I do not know much about this area.

**Review Assessment: Checking Correctness Of Derivations And Theory:**

I did not assess the derivations or theory.

**Review Assessment: Checking Correctness Of Experiments:**

I assessed the sensibility of the experiments.

**Review Assessment: Thoroughness In Paper Reading:**

I read the paper at least twice and used my best judgement in assessing the paper.

---

> ### Author Response · Authors · 2019-11-07
> **Response to Review #2**
>
> Thanks for your feedback.
> The barcode is not the "list of pairs of local minima and their connected saddles".  Please note that we consider specific 1-to-1 correspondence between the set of minima and the set of 1-saddles of "+" type. There can be thousands of local minima and millions of saddle points on paths between fixed pair of minima, even in small dimensions. The barcode of minima extracts the most essential part  from this combinatorially very complex information in the form of simple 1-to-1 correspondence, for each minimum generically there is unique 1-saddle.  The full barcode is the similar decomposition concerning critical points of arbitrary index. To the best of our knowledge this representation of topology of objective functions via canonical 1-to-1 correspondence between minima and saddle points was not studied before in machine learning literature.
>
> We are adding a clarification on this and on other points that you mentioned.
>
> Some general remarks:
>
> Our aim was not just to demonstrate the computation of barcodes of objective functions, but to attract the attention of ML community to this tool. This notion allows to place many different concepts and experiments into a bigger overall picture which improves our understanding and gives useful insights.
>
> Addressing your questions regarding the scalability of our method we would like to mention the following:
>
> 1)All existing topological data analysis packages have scalability issues. Our algorithm permits to raise both the dimension of function’s input, from 4 to 15, and the number of points, from 10^6 to 10^8;
> 2)In the next version of this paper we are adding more experiments with computation of barcodes for loss functions in these dimensions (e.g. related with hyper parameter optimization) which shows usefulness of this approach in other ML problems;
> 3)We currently prepare a sequel paper to this paper where we show the possibility of computation of the barcodes for large-scale modern networks. But even from practical point of view it is important to understand first the behavior of barcodes in simplest examples where all hyper-parameters optimization schemes can be easily turned off.

---

### Official Review · AnonReviewer1 · 2019-10-24
**Official Blind Review #1**

**Rating:** 1

**Review:**

The paper aims to study the topology of loss surfaces of neural networks using tools from algebraic topology. From what I understood, the idea is to effectively (1) take a grid over the parameters of a function (say a parameters of a neural net), (2) evaluate the function at those points, (3) compute sub-levelset persistent homology and (4) study the resulting barcode (for 0/1-dim features) (i.e., the mentioned "canonical form" invariants). Some experiments are presented on extremely simple toy data.

Overall, the paper is very hard to read, as different concepts and terminology appear all over the place without a precise definition (see comments below). Given the problems in the writing of the paper, my assessment is that this idea boils down to computing persistent homology of the sub-levelset filtration of the loss surface sampled at fixed parameter realizations. I do not think that this will be feasible to do, even for small-scale real-world neural networks, simply due to the difficulty of finding a suitable grid, let alone the vast number of function evaluations involved.

The paper is also unclear in many parts. A selection is listed below:

(1) What do you mean by gradient flow? One can define a gradient flow in a linear space X and for a function F: X->R, e.g., as  a smooth curve R->X, such that x'(t) = -\nabla F(x(t)); is that what is meant?

(2) What do you mean by "TDA package"? There are many TDA packages these days (maybe the CRAN TDA package?)

(3) "It was tested in dimensions up to 16 ..." What is meant by dimension here? The dimensionality of the parameter space?

(4) The author's talk about the "minima's barcode" - I have no idea what is meant by that either; the barcode is the result of sub-levelset persistent homology of a function -> it's not associated to a minima.

(5) Is Theorem 2.3. not just a restatement of a theorem from Barannikov '94? At least the proof in the appendix seems to be .

(6) Right before Theorem 2.3., what does the notation F_sC_* mean? This needs to be introduced somewhere.

From my perspective, the whole story revolves around how to compute persistence barcodes from the sub-levelset filtration of the loss surface, obtained from function values taken on a grid over the parameters. The paper devotes quite some time to the introduction of these concepts, but not in a very clear or understandable manner. The experiments are effectively done on toy data, which is fine, but the paper stops at that point. I do not buy the argument that "it is possible to apply it [the method] to large-scale modern neural networks". Without a clear strategy to extend this, or at least some preliminary "larger"-scale results, the paper does not meet the ICLR threshold. The more theoretical part is too convoluted and, from my perspective, just a restatement of earlier results.












**Experience Assessment:**

I have published in this field for several years.

**Review Assessment: Checking Correctness Of Derivations And Theory:**

I assessed the sensibility of the derivations and theory.

**Review Assessment: Checking Correctness Of Experiments:**

I assessed the sensibility of the experiments.

**Review Assessment: Thoroughness In Paper Reading:**

I read the paper at least twice and used my best judgement in assessing the paper.

---

> ### Author Response · Authors · 2019-11-07
> **Response to Review #1**
>
> Thanks for your comments.
> As explained in section 3 we actually do not use any grid. The algorithm for computing barcodes of arbitrary function that we developed works with randomly chosen, or specifically chosen, point cloud in the function’s input. It does not require a grid, thus, it expands the calculations of the barcodes of  functions beyond the dimensions of the input accessible before. To the best of our knowledge such algorithm was not described in literature. If it was, we would be grateful for a reference.
>
> ----------------------------------
> Here are answers to your more specific minor questions:
>
> >  [...]What do you mean by gradient flow? [...]
> The gradient flow is the standard  notion, which if needed can be easily looked up in the cited literature. It is the flow generated by the gradient vector field, the standard vector field used in modern optimization methods.
>
> > [...] What do you mean by "TDA package"?[...]
> The reference to the paper "Introduction to the R package TDA"  is right next to the mentioning of this package.
>
> > [...]Right before Theorem 2.3., what does the notation F_sC_* mean? This needs to be introduced somewhere[...]
> From the text of the paper right before the Theorem 2.3: "...an increasing sequence of subcomplexes (R−filtration) FsC∗ ⊂ FrC∗⊂..."
> so as stated in the paper, FsC∗ ⊂ FrC∗⊂... is indeed an increasing sequence of subcomplexes.
> ---------------------------------------
>
> Some general remarks:
>
> Our aim was not just to demonstrate the computation of barcodes of objective functions, but to attract the attention of ML community to this tool. This notion allows to place many different concepts and experiments into a bigger overall picture which improves our understanding and gives useful insights.
>
> Addressing your questions regarding the scalability of our method we would like to mention the following:
>
> 1)All existing topological data analysis packages have scalability issues. Our algorithm permits to raise both the dimension of function’s input, from 4 to 15, and the number of points, from 10^6 to 10^8;
> 2)In the next version of this paper we are adding more experiments with computation of barcodes for loss functions in these dimensions which shows usefulness of this approach in other ML problems;
> 3)We currently prepare a sequel paper to this paper where we show the possibility of computation of the barcodes for large-scale modern networks. But even from practical point of view it is important to understand first the behavior of barcodes in simplest examples where all hyper-parameters optimization schemes can be easily turned off.

---

### Official Review · AnonReviewer3 · 2019-10-24
**Official Blind Review #3**

**Rating:** 1

**Review:**

This paper introduces the notion of barcodes as a topological invariant of loss surfaces that encodes the "depth" of local minima by associating to each minimum the lowest index-one saddle. An algorithm is presented for the computation of barcodes, and some small-scale experiments are conducted. For very small neural networks, the barcodes are found to live at small loss values, and the authors argue that this suggests it may be hard to get stuck in a suboptimal local minimum.

I believe the concept of barcodes will be new to most members of the ICLR community (at least it was to me), and I appreciate the authors' effort to convey the ideas through multiple definitions in Section 2. I wasn't able to fully appreciate the importance of Definition 3, and Definitions 1 and 2 were tough to digest owing to imprecise language, but I think I got the main point. I was also unable to fully comprehend the definitions of "birth" and "death" in this context. I'd strongly encourage the authors to improve the readability of this section so that non-experts can follow the story.

It seems like the main contribution is a new algorithm for computing barcodes of minima. I am unfamiliar with prior work in this direction, and I was also unable from the paper to infer what the main improvements were relative to the existing algorithms. I'd encourage the authors to state their explicit algorithmic improvements, and to demonstrate empirically that the new algorithm outperforms the prior ones in the expected ways.

The main experiments are on extremely tiny neural networks, presumably owing to computational restrictions. The authors state that "it is possible to apply it to large-scale modern neural networks", but it's not clear to me how that would work or what additional algorithmic improvements (if any) would need to be made in order to do so. I don't think that the results on tiny neural networks have much relevance to practice, so I think the empirical data presented in this paper will have very limited impact. If there were results for practical models, it would be a different story. So I'd encourage the authors to devote additional effort to scaling up the method for use on practical neural network architectures.

Overall, I think there may be some really nice ideas in this paper that could help shape our understanding of neural network loss surfaces, but the current paper does not explore those ideas fully and does not convey them in a sufficiently clear manner. I hope to see an improved version of this paper at a future conference, but I cannot recommend acceptance of this version to ICLR.

**Experience Assessment:**

I do not know much about this area.

**Review Assessment: Checking Correctness Of Derivations And Theory:**

I assessed the sensibility of the derivations and theory.

**Review Assessment: Checking Correctness Of Experiments:**

I assessed the sensibility of the experiments.

**Review Assessment: Thoroughness In Paper Reading:**

I read the paper at least twice and used my best judgement in assessing the paper.

---

> ### Author Response · Authors · 2019-11-07
> **Response to Review #3**
>
> Thank you for the feedback.
> It seems that there is some misunderstanding here concerning the definition of the minima barcodes. The barcode is not quite associated with the intuition behind the notion of the «depth» of local minima. We associate with each minimum not the lowest index one saddle (which can often lie on a path  to a  higher minimum) but the minimal value 1-saddle among the highest points on paths to different minima with smaller value. We are adjusting the overall exposition on this and on other specific points that you mentioned.
>
> Some general remarks:
>
> Our aim was not just to demonstrate the computation of barcodes of objective functions, but to attract the attention of ML community to this tool. This notion allows to place many different concepts and experiments into a bigger overall picture which improves our understanding and gives useful insights.
>
> Addressing your questions regarding the scalability of our method we would like to mention the following:
>
> 1)All existing topological data analysis packages have scalability issues. Our algorithm permits to raise both the dimension of function’s input, from 4 to 15, and the number of points, from 10^6 to 10^8;
> 2)In the next version of this paper we are adding more experiments with computation of barcodes for loss functions in these dimensions  which shows usefulness of this approach in other ML problems;
> 3)We currently prepare a sequel paper to this paper where we show the possibility of computation of the barcodes for large-scale modern networks. But even from practical point of view it is important first to understand the behavior of barcodes in simple examples where all hyper-parameters optimization schemes can be easily turned off.

---

### Decision · Program_Chairs · 2019-12-19

**Decision:**

Reject

**Comment:**

The main concern raised by the reviewers is that the paper is difficult to read and potentially unclear. Therefore, the area chair read the paper, and also found it fairly dense and challenging to read. While there may be important discoveries in the paper, the paper in its current form makes it too difficult to read. Since four reviewers (including the AC) struggled to understand the paper, we believe the presentation of the paper should be improved. In particular, the claims of the paper should be better put into context.